# Unexpected vertical structure of the Saharan Air Layer and giant dust particles during AER-D

Franco Marenco[1], Claire Ryder[2], Victor Estellés[3], Debbie O'Sullivan[1], Jennifer Brooke[1], Luke Orgill[2,4], Gary Lloyd[5], and Martin Gallagher[5]

[1]Met Office, Exeter, United Kingdom
[2]University of Reading, United Kingdom
[3]University of Valencia, Spain
[4]University of Exeter, United Kingdom
[5]University of Manchester, United Kingdom

**Correspondence:** F. Marenco (franco.marenco@metoffice.gov.uk)

> Manuscript with changes highlighted. Reviewer comment number is indicated in a note for each change.

**Abstract.** The Saharan Air Layer (SAL) in the summertime Eastern Atlantic is typically well-mixed and 3–4 km deep, overlying the marine boundary layer (MBL). In this paper, we show experimental evidence that at times a very different structure can be observed. During the AER-D airborne campaign in August 2015, the typical structure described above was observed most of the times, and was associated with a moderate dust content yielding an Aerosol Optical Depth (AOD) of 0.3–0.4 at 355 nm. In an intense event, however, an unprecedented vertical structure was observed close to the Eastern boundary of the basin, displaying an uneven vertical distribution and a very large AOD (1.5–2), with most of the dust in a much lower level than usual (0.3–2 km). Estimated dust concentrations and column loadings for all flights during the campaign spanned 300–5,500 $\mu$g m$^{-3}$ and 0.8–7.5 g m$^{-2}$, respectively. The shortwave direct radiative impact of the intense dust event has been evaluated to be as large as $-260 \pm 30$ and $-120 \pm 15$ W m$^{-2}$ at the surface and top of atmosphere, respectively. We also report the correlation of this event with anomalous lightning activity in the Canary Islands.

In all cases, our measurements detected a broad distribution of aerosol sizes, ranging from $\sim 0.1$ to $\sim 80$ $\mu$m (diameter), thus highlighting the presence of giant particles. Giant dust particles were also found in the MBL. We note that most aerosol models may miss the giant particles due to the fact that they use size bins up to 10–25 $\mu$m. The unusual vertical structure and the giant particles may have implications for dust transport over the Atlantic during intense events, and may affect the estimate of dust deposited to the Ocean. We believe that future campaigns could focus more on events with high aerosol load, and that instrumentation capable of detecting giant particles will be key to dust observations in this part of the world.

# 1 Introduction

The Saharan Air Layer (SAL) is a deep, hot and dry layer of air transported over the tropical Atlantic from the Western African coast, above a cool and moist Marine Boundary Layer (MBL) (Carlson and Prospero, 1972; Karyampudi et al., 1999; Dunion and Velden, 2004). It is generally associated with a mid-level Easterly jet, and it oftens displays a large content of mineral dust from Northern Africa. Using observational data and chemical transport model simulations of dust lifetime, Kok et al. (2017) found that atmospheric dust is substantially coarser than represented in current global climate models, i.e. that the particle size distribution in the models has too many fine particles and too few coarse particles. As coarse dust warms the climate, they also found that the global dust direct radiative effect is likely to be less cooling than estimated by models in a current global aerosol model ensemble, and that following this a net warming of the planet by dust cannot be ruled out.

A number of experiments have been carried out to characterise the properties of mineral dust and the associated meteorology, targeting the African continent (Redelsperger et al., 2006; McConnell et al., 2008; Haywood et al., 2008; Heintzenberg, 2009; Weinzierl et al., 2009; Haywood et al., 2011b; Ryder et al., 2015) and the Atlantic region (Reid and Maring, 2003; Haywood et al., 2003; Chen et al., 2011; Ansmann et al., 2011; Pio et al., 2014; Chouza et al., 2016; Weinzierl et al., 2017; Rittmeister et al., 2017).

According to the conceptual model by Karyampudi et al. (1999), dust is lifted in the Saharan Boundary Layer in the source region, and then transported Westward. This transport mechanism results in a typical vertical structure which sees a dust-laden SAL between $\sim 850$ hPa ($\sim 1.5$ km) and $\sim 500$ hPa ($\sim 6$ km) in the Eastern Atlantic, characterised by a uniform potential temperature. There is then a reduction of the SAL top height, as dust travels across to the Americas, accompanied by a rise of its base. The quantity and size distribution of dust are however not completely understood (Marsham et al., 2008; Laurent et al., 2010). Moreover, the dust vertical distribution affects its lifetime, as free tropospheric particles are generally long lived compared to boundary layer particles which are turbulently mixed down to the surface.

Using airborne dust as a tracer, the main properties of the SAL have been characterised using the spaceborne lidar onboard CALIPSO. Liu et al. (2008) report that African dust is transported across the Atlantic all year long, with strong seasonal variations in the transport pathways (mainly in the free troposphere in summer, and at the low altitudes in winter). They also highlight that transatlantic dust transported at low altitudes is important for all seasons, especially further across the ocean. The Atlantic dusty zones were found to be shifted southward from summer to winter, with a similar southward shift of dust-generating areas over the continent. Tsamalis et al. (2013) present a systematic study of the SAL using CALIPSO over a five year period, showing that the SAL can be identified all year round, with a marked seasonal cycle. They found that the SAL occurs higher in altitude and at Northern latitudes during summer (5–30N and 1–5 km near Africa) than during winter (5S–15N and 0–3 km near Africa), following the Intertropical Convergence Zone (which forms its Southern boundary), and the midlatitude Westerlies (which prevent its further Northward development). Tsamalis et al. (2013) also found that the the vertical structure of the SAL can be described as a single layer, of which the mean altitude and geometric depth decrease towards the Americas.

The spatial distribution of dust over the Atlantic is also addressed in Rittmeister et al. (2017), using lidar observations obtained during a transatlantic cruise from Guadeloupe to Cape Verde. The properties of the SAL were found to be homogeneous from base to top, a result which suggests that particle sedimentation is reduced and/or that upward transport mechanisms are active. These results are further investigated in Ansmann et al. (2017), where the removal of dust in three atmospheric models was found to be too strong, and accompanied with an excess of fine particles.

Senghor et al. (2017) have also studied the spatial and seasonal distribution of mineral aerosols using SeaWiFS, OMI and CALIPSO, and have highlighted a significant change in the vertical distribution at the Western African coastal transition during summer. This transition can be summarised as an uplift of the aerosol over the ocean above an altitude of 1–2 km, as opposed to a profile down to the surface over the continent, and it is not observed in winter. The formation of this elevated dust layer is also studied in Khan et al. (2015), using WRF-Chem simulations at high resolution, together with the observations of the SAMUM-1 campaign: they highlighted the effect of orographic lifting and the interaction of the continental outflow with the sea breeze as key factors. Mortier et al. (2016) have presented nearly a decade lidar observations at M'Bour, Senegal, a site located on the Western African coast at a low latitude (14N): their study also highlights a vertical distribution up to $\sim 5$ km during summertime dust transport events. Lidar measurements at M'Bour are also described by Veselovskii et al. (2016), and a vertical distribution up to $\sim 5$ km during summer was also observed at Cape Verde in May-June 2008 (Tesche et al., 2011).

The vertical distribution of dust goes in pair with its particle size distribution (PSD), as larger particles are thought to be preferentially removed with gravitational settling during transport across the Atlantic. Using results from the SALTRACE field experiment, Gasteiger et al. (2017) investigate the possibility that vertical mixing may occur in the SAL, driven by the absorption of sunlight by dust in daytime. This hypothesis is driven, in particular, by the observed lack of vertical change in the PSD, and the presence of large particles near the top of the SAL.

Here we present results from the AERosol properties - Dust (AER-D) airborne campaign, carried out during August 2015 over the Eastern tropical Atlantic. This campaign was aimed at characterising mineral dust, for the benefit of validating modelling and remote sensing methods, and for improving the knowledge on optical and microphysical properties. Our measurements of the particle size distribution in the coarse and giant mode during summer, and the characterisation of an outbreak with large aerosol optical depth (AOD) contribute with a new perspective to the knowledge of airborne mineral dust. The AER-D observations bring an important insight on mineral dust transported over the basin, and provide a wealth of data that is further analysed in variety of studies (Liu et al., 2018; Price et al., 2018; Ryder et al., 2018; O'Sullivan et al., 2018; Estellés et al., 2018).

## 2 Research flights

Between 6 and 25 August 2015, the Facility for Airborne Atmospheric Measurements (FAAM) research aircraft was based in Praia, Cape Verde, in the island of Santiago (14°57′N, 23°29′W). AER-D was conducted simultaneously with ICE-D (Ice in Clouds Experiment – Dust), aimed at studying dust-cloud interactions and the evolution of towering cumulus clouds due to the role of dust as ice nuclei (Liu et al., 2018). In addition, the Sunphotometer Airborne Validation EXperiment in Dust

(SAVEX-D) was also carried out, and is treated here as a component of AER-D. Sixteen research flights were carried out between AER-D and ICE-D. Here we present data from six AER-D flights (see Table 1), representing nearly 28 h of flight time. All flights were above the Atlantic Ocean, off the West coast of Africa, and they targeted studies on the mineral dust transported within the Saharan Air Layer. The range of conditions was varied, and dust loadings from moderate to very high

were encountered. The area covered is between Cape Verde and the Canary Islands.

The aircraft was equipped with in situ and remote sensing equipment, as described in the following sections. Forecasts of Saharan dust and cloud cover from the Met Office Unified Model (Walters et al., 2011; Mulcahy et al., 2014), as well as from the Copernicus Atmosphere Monitoring Service based at the European Centre for Medium Range Weather Forecasts (Morcrette et al., 2009; Benedetti et al., 2009), were used to plan the research flights. Dust outbreak events could be predicted

several days in advance, and detailed guidance was provided in real time using MSG-SEVIRI satellite imagery products, enabling the flight plans to be adapted just before or during each mission.

The meteorology of the campaign has been analysed in Liu et al. (2018), where two regimes have been identified. In the first part of the campaign, until the 14 August, winds in the SAL above Cape Verde were found to be slack ($< 10$ m s$^{-1}$, and with an oscillating direction between NE and SE). Instead, from the 15 August onwards the winds in the SAL above Cape Verde

were stronger ($> 10$ m s$^{-1}$ and from a marked E direction). This change of regime was associated with the migration of a high at 700 hPa towards the SE, from Western to Central Algeria, leading to a more intense easterly jet. A slacker NE flow ($\sim 5$ m s$^{-1}$) was found in the MBL, controlled by the trade winds.

Three types of research flights have been carried out:

1. Remote sensing (RS) flights, designed mainly to provide mapping of the dust layer using the airborne lidar (three flights),

and accompanied with additional measurements using dropsondes and radiometers. These flights typically aimed for the heavy dust outbreaks off the African coast, and travelled a long distance at high altitude; limited in-situ sampling at lower altitudes was also conducted. The goal of these flights was to provide a small but valuable dataset, useful for the validation of satellite retrievals, model forecasts, and dust data assimilation schemes.

2. Validation of the retrievals of aerosol microphysical properties from two types of ground-based sunphotometer (PREDE / 

SKYNET and CIMEL / AERONET) under the SAVEX-D project (two flights). For these flights, we sampled a limited geographic area, as near as possible to a ground-based site equipped with sunphotometers, and we aimed to fully characterise the atmospheric column by sampling at several different altitude levels.

3. The validation of products from the Cloud-Aerosol Transport System lidar (CATS; Yorks et al., 2016) on-board the International Space Station (ISS) (one flight). This type of flight was coordinated with the predicted track of the ISS, and

a flight pattern similar to SAVEX-D was adopted (limited geographical area, and sampling at several different altitudes).

Note that all times reported in the current paper are UTC.

## 3 Vertical structure

Fig. 1 displays the vertical profiles of aerosol extinction coefficient at 355 nm during the high altitude transects. They have been obtained with the on-board elastic backscatter lidar, and they have been evaluated using previously published methods (Marenco et al., 2011, 2014; Marenco, 2013; Marenco et al., 2016). The vertical resolution of the processed dataset is 45 m and the integration time is 1 min, corresponding to a $\sim 9$ km footprint. The AER-D lidar measurements and their uncertainties are described in more detail in O'Sullivan et al. (2018). In brief, the lidar inversion is based on a double iteration. In the first iteration, a campaign-mean lidar ratio (extinction-to-backscatter ratio) is computed, which ensures consistency of the lidar signals with the layers identified as aerosol-free. Then, the data analysis is re-iterated using this lidar ratio ($54 \pm 8$ sr), and using the slope-Fernald approach, based on the use of a far-end reference within the bottom portion of the aerosol layer (see Marenco, 2013 for details).

In most cases, a deep dust layer is identified, with base at 1–2 km and top at 5–6 km altitude, above a MBL also displaying a significant aerosol content (see Fig. 1a). As expected with aerosol fields, there are day-to-day variations; however the main properties are consistent: the AOD is in the 0.3–0.4 range, the aerosol extinction coefficient is of the order of 100–200 $\text{Mm}^{-1}$, and the depth of the elevated dust layer is 3–4 km. This observed structure is in agreement with expectations from the conventional model for Saharan dust transport over the Atlantic (Carlson and Prospero, 1972; Karyampudi et al., 1999). We shall classify this typical condition as "moderate dust".

During flights B923 and B924, however, a different vertical structure was observed near the Western African coast (Fig. 1b). The AOD was 1.5–2, the MBL was compressed with a top at $\sim 0.3$ km, and the dust extended from the top of the MBL up to a layer top at $\sim 5.3$ km. The highest loadings were found at low altitudes (1–2 km), with extinction coefficients in excess of 1,000 $\text{Mm}^{-1}$, and we shall denote this condition as "heavy dust" and/or "anomalous structure". Temperature, moisture and wind profiles (not shown here) suggest that the upper dust layer (2–5 km) was well mixed (constant potential temperature, $\theta$), and characterised by a moderate ESE flow. On the other hand, the lower layer (0.3–2 km), where the intense dust concentration was found, was stable (increasing $\theta$ with height and constant temperature), and displayed a northeasterly wind (Ryder et al., 2018). We believe that this observed vertical distribution of dust, with a double layer and a large concentration very low above a compressed MBL, reveals a surprising exceptional structure, not previously encountered during measurements of the SAL over the Atlantic Ocean, and it is particularly interesting since it coincides with a large AOD event. This observation shows that the anomalous structure can exist in the near range for Saharan dust transport across the Atlantic (100–300 km from the coast); however nothing can be inferred for longer transport distances. Moreover, nothing can be inferred concerning the frequency of occurrence of the anomalous structure, but the fact that it wasn't reported before may suggest that it is sporadic and limited to heavy dust outbreaks.

## 4 Particle size distribution

The layers with the largest concentrations, identified by lidar or during aircraft ascent and descent, have been sampled in situ with the aircraft instruments. Fig. 2 shows the particle size distributions (PSDs) obtained during 19 straight and level runs

(SLRs) sampled during the AER-D flights. We have subdivided the PSDs into marine boundary layer (MBL), moderate dust loading, and heavy dust loading, based on the concentrations observed. The heavy dust PSD was collected in flight B924 (Run 5 at 1,000 m altitude), at the same location where the lidar highlighted the heavy dust condition, whereas the moderate dust PSDs refers to all the other samples, and hence corresponds to the cases clustered as moderate dust in the previous section. The PSDs shown here have been obtained using three wing-mounted probes which, when combined together, are capable of sampling the spectrum between 0.1 and 1000 $\mu$m (diameter): Passive Cavity Aerosol Spectrometer Probe (PCASP), Cloud Droplet Probe (CDP), and Two-Dimensional Stereo Probe (2DS) (Knollenberg, 1981; Liu et al., 1992; Lance et al., 2010; Crosier et al., 2011; Rosenberg et al., 2012; Ryder et al., 2013). Calibration of the PCASP was done before and after the campaign, whereas the CDP was also calibrated before most flights. The CDP size resolution was enhanced at the smaller end of the spectrum by using custom settings and making use of the particle-by-particle data. For both probes, particle spectra have been processed for an assumed refractive index of dust of $1.53 - 0.001i$, thus correcting for the bin ranges calibrated using polysterene latex spheres, and the first bin has been discarded due to its undefined lower edge. The 2DS is a shadowing probe with 10 $\mu$m resolution, and it does not rely on refractive index to infer particle size. The AER-D particle size distributions and the instruments are described in detail in Ryder et al. (2018).

We note that authors in the geological sciences often consider that 62.5–2,000 $\mu$m particles are "sand" as opposed to "dust." Here, however, we will use the term "dust" for the particles that we observed, adhering to a terminology in use in the atmospheric sciences, where "dust" is considered to be suspended material transported by the wind (Kok et al., 2012).

The volume PSDs are dominated by a very broad coarse mode centred at 5–6 $\mu$m (diameter), which is also where the volume distributions peak. Giant particles (diameter 20–80 $\mu$m) were detected for 75% of the samples. Runs in the MBL exhibit a clearly pronounced fine mode, peaking at $\sim 0.2$ $\mu$m diameter, whereas for the dust layers the fine mode peaks at 0.25–0.3 $\mu$m diameter and is less marked. For 25% of the in-dust SLRs, a distinct fine mode is not observed. As expected, the largest concentrations were encountered in the heavy dust case (flight B924). Somewhat surprisingly, however, some of the SLRs in the MBL exhibit a concentration of giant particles similar to the heavy dust case.

For the samples taken in dust, giant particles do not represent a separate mode of the size-distribution, but rather an extension and broadening of the coarse mode. Contrastingly, for two of the MBL samples the giant particles appear as a separate mode with diameter between $\sim 15$ and $\sim 80$ $\mu$m, showing a broad peak at 20–40 $\mu$m.

Table 1 displays the effective diameter ($D_{\text{eff}}$), derived from the PSDs; the SLRs in dust exhibited a $D_{\text{eff}}$ of 3.6–4.7 $\mu$m, whereas in the MBL $D_{\text{eff}}$ was 3.4–5.5 $\mu$m, thus exhibiting a larger variability from flight to flight.

The in situ measurements from AER-D are discussed in more detail in Ryder et al. (2018). Moreover, Liu et al. (2018) describes the measurements of hematite content of dust during ICE-D and AER-D, and their dependency on dust age.

## 5    Exceptional dust event

On 12 August a major outbreak of dust occurred West of the African continent: see the Meteosat Second Generation (MSG) image shown in Fig. 3. From model predictions, significant dust concentrations were expected between Western Sahara and the

Canary Islands. A targeted mission was planned, consisting of a double flight on a single day. Flight B923 was a three-hour high altitude mapping flight held in the morning, from Praia to Fuerteventura, and it was followed by flight B924 in the afternoon. The latter was a full five-hour scientific flight, and in addition to dust mapping it also allowed for a 1.5 h-long descent in the dust layers, enabling the in situ sampling at two altitude levels, 30 m and 1,000 m above sea-level. Fig. 3 shows the flight tracks, and a red circle is used to highlight the area of the in situ sampling: it can be seen from the underlying satellite image that the latter area is ideally located at the leading edge of the "dust front".

Fig. 4 shows the vertical structure of the atmosphere, as revealed by the lidar during the three high altitude transects. The plots reveal both the aerosol and the cloud fields, and the location of the vertical profiles shown in Fig. 1 is indicated with red boxes. This figure clearly shows the contrast between the pre-front SAL (moderate dust conditions) and the much more intense dust loading in the post-front SAL (heavy dust). The pre-front SAL is on the left/South and the post-front on the right/North. It also shows well that the front advanced towards the South, and that its leading edge featured a very intense dust layer at low altitude (1–2 km).

Fig. 4a represents the first flight, heading Northwards. We initially observed a shallow SAL at 16N, identified through the presence of dust, and positioned above the marine stratocumulus deck. As we continued, the SAL deepened until we reached 20N: there, we overflew a high cloud deck, situated at the top of the SAL and obscuring the layer below. The band of clouds is clearly identified in the 10:30UTC MSG imagery (not shown here), and it runs in a Southeast-Northwest direction, crossing our track orthogonally. The image in Fig. 3 refers to the afternoon (16:30UTC), when the band of clouds had mainly disappeared, although some scattered clouds remain (it still suggests the location of the band we encountered in the morning). At 21.2N, beyond this band of clouds, we continued overflying a SAL extending between 1–2 and $\sim 5$ km, exhibiting a moderate AOD ($\sim 0.34$) and a moderate extinction coefficient ($\sim 100$ $\mathrm{Mm^{-1}}$). No further clouds were encountered during this flight. As we reached a point at 24N,17W, the aerosol load and its vertical profile changed: the AOD was observed to increase to $\sim 1.5$ and the intensity of the extinction coefficient increased markedly at all altitudes. At its leading (Southward) edge, this "dust front" exhibited a very intense extinction coefficient, especially in its lower layers ($\gtrsim 1,000$ $\mathrm{Mm^{-1}}$ at around $\sim 1$ km, and of the order of 200–400 $\mathrm{Mm^{-1}}$ up to altitudes of $\sim 5$ km).

On the return flight (Fig. 4b+c), the "dust front" boundary was crossed again, and similar features were observed once more. The extinction coefficient at its peak intensity had risen to $\sim 1,500$ $\mathrm{Mm^{-1}}$, and the AOD reached 2. The boundary between the heavy dust and the moderate dust had advanced South and West: this Southward motion is evident when considering the succession of panels (a), (b) and (c) in Fig. 4.

Between 15:26 and 16:48UTC a descent of the aircraft into the "front" was made, at around 24N,18.2W. The in situ sampling levels were initially identified during flight at high altitude, using the real-time lidar display, and refined during the descent using our nephelometer. The nephelometer showed that the dust layer extended from 0.3 km to 5.2 km, with a sharp boundary at its base; we consider the air below this base to be in the MBL. An intense layer was observed between 0.85 and 1.4 km, with a 550 nm extinction coefficient of 2,100 $\mathrm{Mm^{-1}}$ as determined by the nephelometer in combination with a Particle Soot Absorption Photometer, whereas above 2.6 km the extinction coefficient was only 140 $\mathrm{Mm^{-1}}$. Extinction in the MBL was 100–150 $\mathrm{Mm^{-1}}$. During the SLR at 1,000 m, particles with diameter up to 80 $\mu$m were observed (green line, Fig. 2), with an

increased particle concentration for all sizes. In comparison, the measurements in the MBL (dashed blue line in Fig. 2) showed a much smaller number of particles in the 0.25–40 $\mu$m diameter range, whereas similar particle numbers were observed below $\sim 0.25$ $\mu$m and above $\sim 40$ $\mu$m. The most striking feature is the fact that the size and concentration of giant particles in the MBL is very similar to the ones observed in the dust layer.

MSG "dust RGB" imagery permitted us to track the origin of this dust (see Ryder et al., 2018). The dust that we sampled at 24N, 18.2W, at the edge of the dust "front", seems to have been uplifted in Northern Mali two days earlier, during a haboob generated by a mesoscale convective system, and to have travelled 2,000 km towards the WNW towards the location where we sampled it. However, we also found two other dust uplifting events, that may have contributed to the wider dust outbreak off the African coast on that day: one of them happened in Central Algeria (2,000 km, 2 days transport) and the other one in

Northern Niger (3,000 km, 3 days transport). The CATS spaceborne lidar detected dust that can be ascribed to these events, between 0:54 and 0:59 UTC on 11 August (not shown here). The 1064 nm total attenuated backscatter image from CATS also revealed a peculiar vertical distribution, with a layer in the 2–6 km altitude range and another closer to the surface, not too dissimilar from our anomalous structure.

## 6   Shortwave radiative effect of the exceptional dust event.

Upward and downward facing broadband pyranometers measured the downwelling and upwelling solar irradiance between 0.3 to 3 $\mu$m during straight and level runs. They have been used in conjunction with a radiative transfer model to calculate the surface and top of atmosphere (TOA) solar direct radiative effect of the exceptional dust event measured during flight B924. We used the established methodology of Haywood et al. (2001, 2003, 2011a). The shortwave upwelling and downwelling irradiance with aerosol present, $SWU_{aer}$ and $SWD_{aer}$, was measured by the pyranometers. The shortwave upwelling and

downwelling irradiance with no aerosol present (clear sky), $SWU_{cs}$ and $SWD_{cs}$, was calculated from a radiative transfer model.

The direct radiative effect on SWU at the TOA has been estimated as follows, using measurements during a high level straight level run above the dust layer:

$$DRE_{SW,TOA} = SWU_{cs} - SWU_{aer} \tag{1}$$

At the surface, pyranometer irradiance measurements from a low level straight level run below the layer were similarly used to determine the direct solar radiative effect of the dust at the surface, as follows:

$$DRE_{SW,SURF} = SWD_{aer} - SWD_{cs} \tag{2}$$

For model calculations, the Suite Of Community RAdiative Transfer codes based on the SOCRATES) model was used (Edwards and Slingo, 1996; Randles et al., 2013), configured to use two streams and 6 spectral bands to represent the spectral

range of the aircraft pyranometers. Surface albedo values were derived from pyranometer observations during the low level run, and solar properties were accounted for. Vertical profiles of atmospheric temperature and composition were taken from a

standard tropical atmosphere (McClatchey et al., 1972), but replaced by aircraft in-situ measurements for temperature, water vapour and ozone for the layers sampled during ascent and descent (19–6,000 m). A range of irradiances was calculated, where surface albedo was varied between 0.03–0.06 (the range measured during the low level run), solar zenith angle was varied in a range corresponding to the times of each relevant run, and the meteorology was varied between the values measured during aircraft descent and ascent. In this way, we account for the uncertainty due to spatial variability in the input parameters.

Pyranometer data have been corrected to account for the pitch and roll of the aircraft as a function of time, using offsets of the instruments relative to the aircraft fuselage of $-4.6°$ and $+0.9°$, respectively, determined from a series of dedicated "box pattern" and "pirouette" manoeuvres during the campaign (see McConnell, 2009; Haywood et al., 2011a). An uncertainty of 5.5% was adopted for the pyranometer measurements based on McConnell (2009), and this includes a contribution of the levelling corrections.

Fig. 5 shows the low level (30 m) and high altitude (6–6.5 km) irradiance measurements, as well as the AOD from the aircraft lidar (the latter was determined during the high level runs). The Southern end of these plots corresponds to moderate dust conditions, whereas the Northern end to heavy dust conditions. Note that the two high level runs considered here overflew the same segment of the low level measurements, respectively beforehand and afterwards.

In Fig. 5a, it can be seen that the $SWD_{aer}$ measured at low level (run 4) dropped by 90 W m$^{-2}$, from around 610 W m$^{-2}$ to 520 W m$^{-2}$ at the Northern end of the run, i.e. underneath the dust "front" (note the rise in AOD from $< 0.7$ to 1–1.5). The resulting surface $DRE_{SW,SURF}$ is shown in Fig. 5b, and it changes from $-170$ W m$^{-2}$ in the South to the larger negative value of $-260 \pm 30$ W m$^{-2}$ in the North, under the dust "front."

SWU irradiance measurements above the dust layer are shown in Fig. 5c, for runs 3 and 6, before and after the descent into the dust layer, respectively. During the time between these two runs (1.5 h), the solar zenith angle increased from $30°$ to $51°$, and the dust "front" moved southwards. The increase in SWU in the location of the larger AOD can be seen in Fig. 5c, with SWU increasing by around 50 W m$^{-2}$ and 35 W m$^{-2}$ for runs 3 and 6, respectively. Fig. 5d shows the TOA $DRE_{SW,TOA}$, which was more negative over the intense dust loading, changing from around $-50$ W m$^{-2}$ to $-120$ W m$^{-2}$ (run 3) and from $-65$ Wm$^{-2}$ to $-100$ Wm$^{-2}$ (run 6). For the larger solar zenith angle (run 6), the $DRE_{SW,TOA}$ is more negative due to the greater backscatter fraction, although the lower incoming solar radiation means that the change in $DRE_{SW,TOA}$ from ahead of the dust "front" to over it is smaller in magnitude.

$DRE_{SW,TOA}$ values have been combined with the AOD measurements to determine the dust shortwave radiative efficiency. Values computed for the TOA vary between $-50$ and $-95$ W m$^{-2}\tau^{-1}$. Run 3 shows no latitudinal dependence, while run 6 shows a clear trend of most negative values, with $-88$ W m$^{-2}\tau^{-1}$ in the South and $-51$ W m$^{-2}\tau^{-1}$ in the North. This difference between the two runs can be attributed to the complexity in relating the pyranometer measurements, with a near-hemispherical view, to the lidar, measuring vertically downwards, and how this relationship may change with varying solar zenith angle due the phase function of the dust. Surface radiative efficiencies have also been estimated (but note that the AOD was not measured concurrently to the irradiance): taking AOD and $DRE_{SW,SURF}$ values at the beginning and end of the low level run gives an approximation of $-230$ and $-170$ W m$^{-2}\tau^{-1}$ ahead of and under the dust "front," respectively.

## 7 Conclusions

The AER-D campaign encountered some unprecedented conditions associated with an outbreak of dust slightly downstream of continental Africa. Typical and exceptional properties of the layer have been documented during the field deployment, and the concentration of giant particles has been measured. We believe that these results contribute to reinforce the evidence in favour of a presence of giant and coarse dust particles in the Eastern Atlantic, which may currently be missed in most aerosol models (see e.g. Table 1 of Huneeus et al., 2011). The dust size distribution strongly controls the radiative impact of the aerosols, as well as their interactions with clouds. The size of particles also controls how far downwind they travel, and thus their ability to impact biogeochemistry downwind of the source region (Mahowald et al., 2014).

The magnitude of the dust direct and indirect radiative effects is still uncertain, and it remains unclear whether atmospheric dust has a net warming or cooling effect on global climate. The dust particle size distribution has a role to play in this because fine particles predominantly scatter solar radiation (cooling effect), whereas for coarse particles absorption of solar and thermal radiation plays a larger role (warming). By applying experimental constrains to the global dust abundance and particle size, Kok et al. (2017) conclude that the global dust direct radiative effect is likely to be less cooling than estimated currently by models, and that a net warming effect is not to be ruled out. Our finding of very large particles in the Eastern Atlantic, both in the SAL and in the MBL, supports these conclusions.

Our observations of the SAL vertical structure and particle size distribution have been classified into two clusters: moderate dust (encountered most of the times) and heavy dust (encountered on 12 August, North of 24N).

In general, the SAL geometry in the Eastern Atlantic in the season considered was one of a deep layer between 1–2 km and 5–6 km, with an extinction coefficient at 100-200 $Mm^{-1}$ and an AOD of the order of 0.3–0.4 (moderate dust), in agreement with the findings of previous studies such as e.g. Tsamalis et al. (2013); Rittmeister et al. (2017). Using the specific extinction estimated by Ryder et al. (2018) (0.27–0.35 $m^2\,g^{-1}$), these extinctions and AODs translate into estimated dust concentrations around 300–700 $\mu g\,m^{-3}$, and column loadings of 0.8–1.5 $g\,m^{-2}$. The latter figure is comparable with the estimate obtained from the insitu data of about 1 $g\,m^{-2}$ (Ryder et al., 2018). In situ measurements displayed comparable particle sizes with the ones reported for the SAMUM-2 campaign (Ansmann et al., 2011), and highlighted a significant contribution from giant particles.

On 12 August 2015 we observed a heavy dust outbreak, uplifted in Northwestern Africa two–three days earlier, which featured an AOD of 1.5–2, and a particularly intense layer at 1–2 km, where the extinction coefficient was 1,000–1,500 $Mm^{-1}$. This corresponds to estimated concentrations between 3,000 and 5,500 $\mu g\,m^{-3}$, and column dust loadings of 4–7.5 $g\,m^{-2}$. The latter figure is slightly larger than the estimate from the in-situ measurements (3–6 $g\,m^{-2}$, Ryder et al., 2018). The vertical distribution for this event is unprecedented over the Atlantic, and our combination of remote sensing and in situ measurements yields a unique insight into the properties of this event. Note also that when observing this layer, we are pushing the observation by lidar to its limits, as extinction of the signal is large. The PSD for this intense layer showed the largest numbers and the largest particles, and it exhibited only one very broad mode. Giant particles up to $\sim 80$ $\mu m$ (diameter) were detected, and giant particles with the same PSD signature were also found in the underlying shallow MBL, despite a much less intense

coarse mode. In principle, the giant particles in the MBL could be either dust particles depositing from above, or sea spray (or a combination of both). However, as documented in Ryder et al. (2018), the analysis of filter samples collected during the research flights suggests that giant particles observed in the MBL during the AER-D flights are mineral dust, thus highlighting what is likely the effect of dry deposition.

5 The unusual heavy dust event was associated with a large direct solar radiative effect, reaching $-260 \pm 30$ and $-120 \pm 15$ W m$^{-2}$ at the surface and top of atmosphere (TOA), respectively. Moreover, we registered a rapid spatial variation across the dust front, of the order of 90 and 35–50 W m$^{-2}$, respectively. Such perturbations to the radiative budget are significant, and to the authors' knowledge it is the first time that such large values have been measured over ocean, in combination with dust in-situ, vertically resolved properties. Slingo et al. (2006) documented a major dust storm over Niger, with extremely high

10 AOD peaking 3–4, and subsequent midday solar direct radiative effect of $-100$ W m$^{-2}$ at the TOA and $-250$ W m$^{-2}$ at the surface. Their surface and TOA flux changes were of a similar magnitude to those measured over the dust "front" discussed here; however, the change per AOD unit that we observed is larger, and this can be explained with the low ocean albedo. Haywood et al. (2011a) performed similar estimates, for a moderate AOD of 0.26 over the tropical Eastern Atlantic, finding a smaller radiative effect of $-47$ and $-33$ W m$^{-2}$ at the surface and at the TOA, respectively.

15 Moreover, the intense dust event was followed by anomalous electric discharges. On 12 and 13 August 2015, the Spanish Meteorological State Agency's Lightning Detection Network (Red de Detección de Rayos de AEMET, REDRA) recorded nearly 6,000 lightning strikes in the Canary Islands region (Prats et al., 2018). The lightning activity began at around 16:00 on the 12th, and reached its maximum intensity between 19:00 and 02:00 during the night, followed by additional strikes on the second day. Meteosat Second Generation imagery shows the formation of convective cells over the area during the night,

20 which are likely connected to the electrical activity. This lightning density was described as extraordinary, because in 12 years of operations the network has only observed a limited number of events (23 days) of a comparable intensity, all of which were during the autumn and winter seasons.

 It is beyond the scope of this paper to establish whether there is a causal link between the dust event and the electric discharges. Note that the association of aerosols with lightning is still a relatively new field of science, where the understanding

25 of the underlying processes is still weak. Yuan et al. (2011), showed an increased lightning activity East of the Philippines, associated with secondary sulphates that formed following the eruption of Mt. Anatahan, Mariana Islands, and demonstrated that aerosols increase lightning activity in tropical regions through modification of cloud microphysics. Moreover, Thornton et al. (2017) showed that lightning density can be twice as large over shipping lanes, thus reinforcing the hypothesis that aerosols may lead to a microphysical enhancement of convection and of storm electrification. Similar observations where also reported

30 over the Mediterranean region, where moreover a correlation was found with the aerosol extinction coefficient between $\sim 1$ and $\sim 3$ km, and the aerosols mostly associated to lightning events over this area were identified as being dust and smoke (Proestakis et al., 2016). As a word of caution, however, we have to mention that observations over the Amazon basin seem to cast doubt on a primary role for aerosols in enhancing cloud electrification (Williams et al., 2002).

 Our results trigger some questions and the need for further research. (1) How frequent is an anomalous vertical structure

35 like the one we encountered? We are not aware of other measurements in intense outbreaks 100–300 km off the coast of West

Africa, and we believe that further field deployments targeting this region could offer insights. An analysis of past satellite observations may also help assess this. (2) How long lived are giant particles like the ones we observed? Only few airborne campaigns have been able to measure the full particle size spectra, and little information is available: once again, this may be addressed in future campaigns. (3) What may have happened subsequently to the considerable amount of dust that we encountered at an altitude of 1–2 km during our intense case? Has it mixed through the SAL, was it deposited rapidly into the ocean (as could be suggested by the finding of giant particles in the underlying MBL), or did it keep its distinct identity downstream? (4) Does dust play a role in triggering convection and lightning events over the Atlantic, like the one that affected the Canary Islands on 12–13 August 2015?

Answering such questions may be vital for a better understanding of the aerosol processes driving deposition, transport, and aerosol-cloud interactions, and hence for the improvement of models and assessing the climate impact of dust. Further observations are needed, focusing on the evolution of aerosol properties during transport across the Atlantic, and a coordinated experiment on both sides of the basin could be a means to achieve this.

*Data availability.* The FAAM aircraft datasets for the ICE-D and AER-D campaigns can be requested from the British Atmospheric Data Centre, Centre for Environmental Data Analysis (http://data.ceda.ac.uk/badc/ice-d/).

*Acknowledgements.* Airborne data were obtained using the BAe-146-301 Atmospheric Research Aircraft operated by Directflight Ltd and managed by the FAAM, which is a joint entity of NERC and the Met Office. The staff of the Met Office, the Universities of Leeds, Manchester and Hertsfordshire, FAAM, Direct Flight, Avalon Engineering and BAE Systems are thanked for their dedication in making the ICE-D and AER-D campaigns a success. Claire Ryder was funded by NERC grant NE/M018288/1. SAVEX-D was possible thanks to EUFAR TNA (European Union Seventh Framework Programme grant agreement 312609) and projects PROMETEUII/2014/058 and GV/2014/046 from the Valencia Autonomous Government, and CGL2015-70432-R from the Spanish Ministry of Economy and Competitiveness—European Regional Development Fund. We thank CAMS/ECMWF for providing model products in support of the ICE-D and AER-D campaigns, and Eumetsat for providing imagery from MSG-SEVIRI in real time.

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

**Table 1.** The AER-D research flights.

| Flight | Date | Time | Latitude | Longitude | SAL $D_{\text{eff}}$ ($\mu$m) | MBL $D_{\text{eff}}$ ($\mu$m) | Flight type[a] |
|--------|------|------|----------|-----------|------------------------------|------------------------------|----------------|
| B920 | 7 Aug | 14:04–18:45 | 14.6–17.9N | 21.0–23.6W | 3.9–4.0 | 3.4 | CATS |
| B923 | 12 Aug | 9:02–12:10 | 14.9–28.4N | 13.7–23.5W | N/A | N/A | RS[b] |
| B924 | 12 Aug | 13:59–19:00 | 14.9–28.4N | 14.3–23.5W | 4.3 | 4.5 | RS[b] |
| B928 | 16 Aug | 14:00–19:08 | 14.0–16.2N | 23.4–24.1W | 3.6–4.7 | 5.2 | SAVEX-D |
| B932 | 20 Aug | 8:58–13:52 | 14.6–20.5N | 18.6–23.5W | 3.7–4.3 | 5.5 | RS |
| B934 | 25 Aug | 13:58–18:49 | 14.8–17.7N | 23.0–23.5W | 3.8–4.0 | 4.4 | SAVEX-D |

[a]RS: airborne Remote Sensing (with limited in-situ sampling);

SAVEX-D: Sunphotometer Airborne Validation EXperiment in Dust;

CATS: underflight of the CATS lidar on the ISS.

[b]B923: from Praia ($14°57'$N, $23°29'$W) to Fuerteventura ($28°27'$N, $13°52'$W);

B924: from Fuerteventura to Praia.

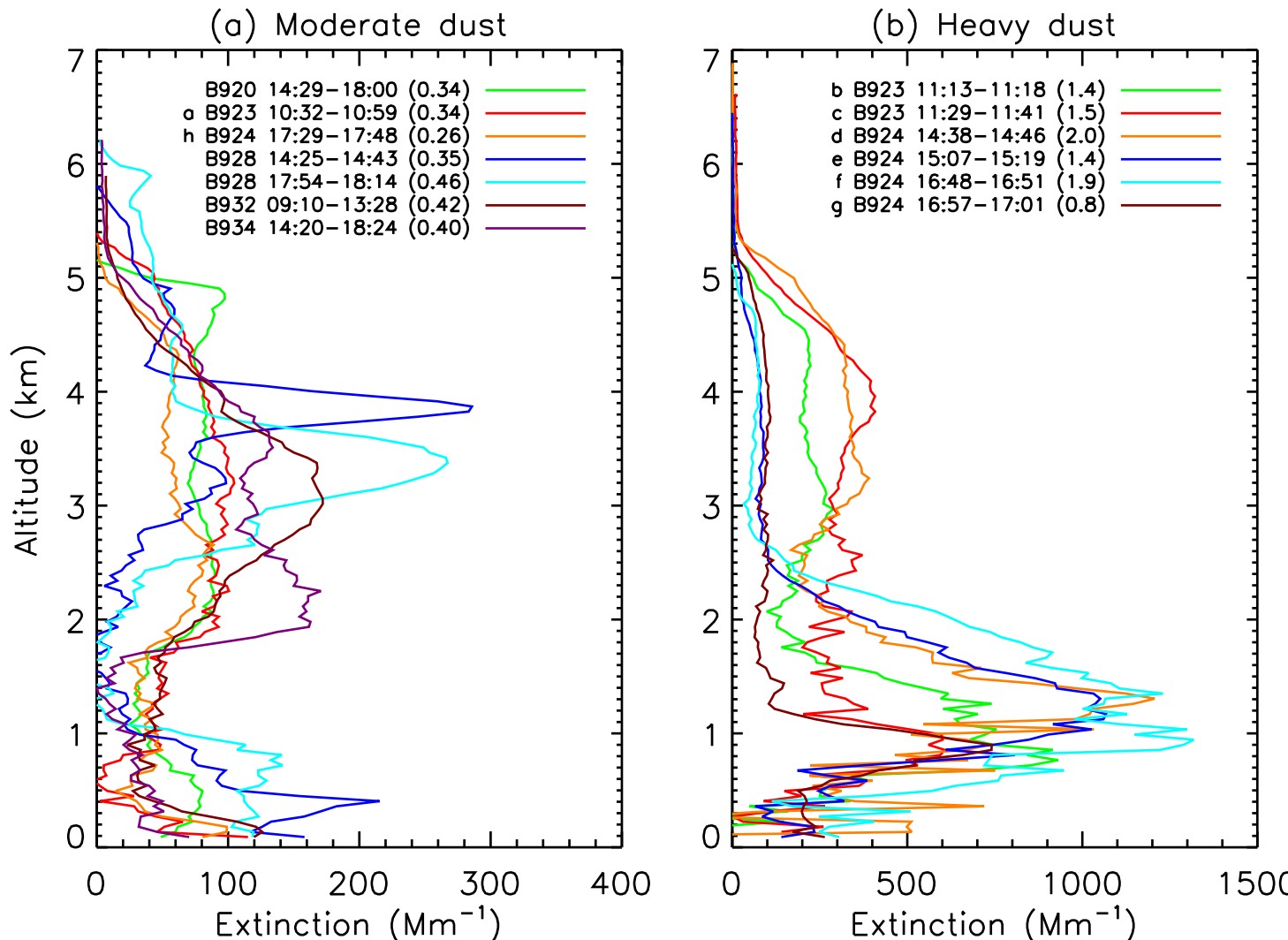

**Figure 1.** Vertical profiles of aerosol extinction coefficient measured by airborne lidar, identified by flight number and time (UTC). The number in parentheses is the AOD resulting from the lidar profile. For flights B923 and B924, a letter (a-h) identifies the transects in Fig. 4 used to compute the profiles shown in this figure. Panel (a) refers to the moderate dust conditions, most often encountered during the campaign, whereas panel (b) refers to the heavy dust loads encountered on 12 August.

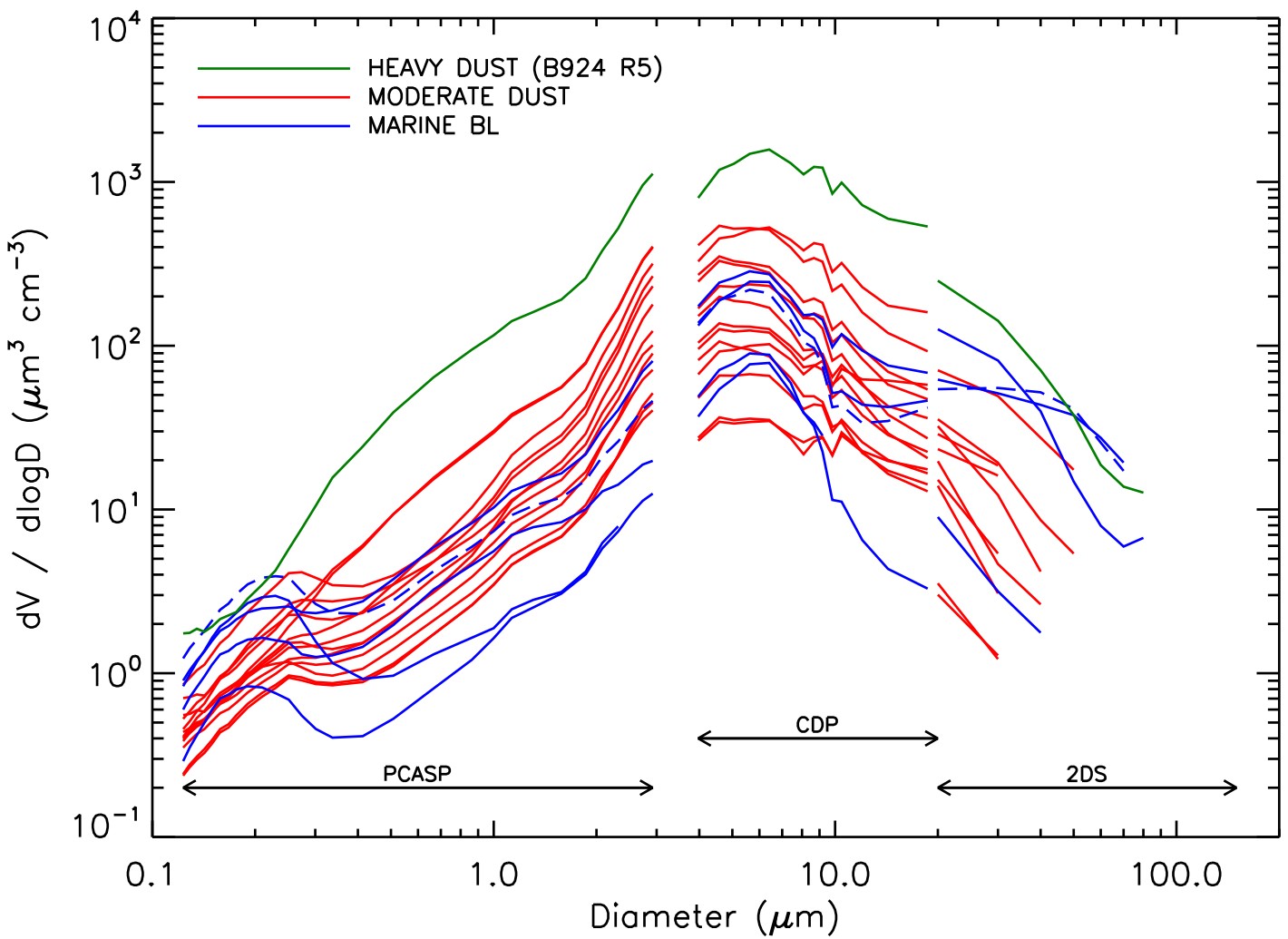

**Figure 2.** Particles size-distributions obtained during in situ straight level runs using the PCASP (0.1–3 $\mu$m), CDP (4–20 $\mu$m), and 2DS (>20 $\mu$m). Blue: MBL; red: SAL, moderate dust loading; dark green: SAL, heavy dust loading (flight B924 run 5); dashed blue: MBL under heavy dust layer (flight B924 run 4).

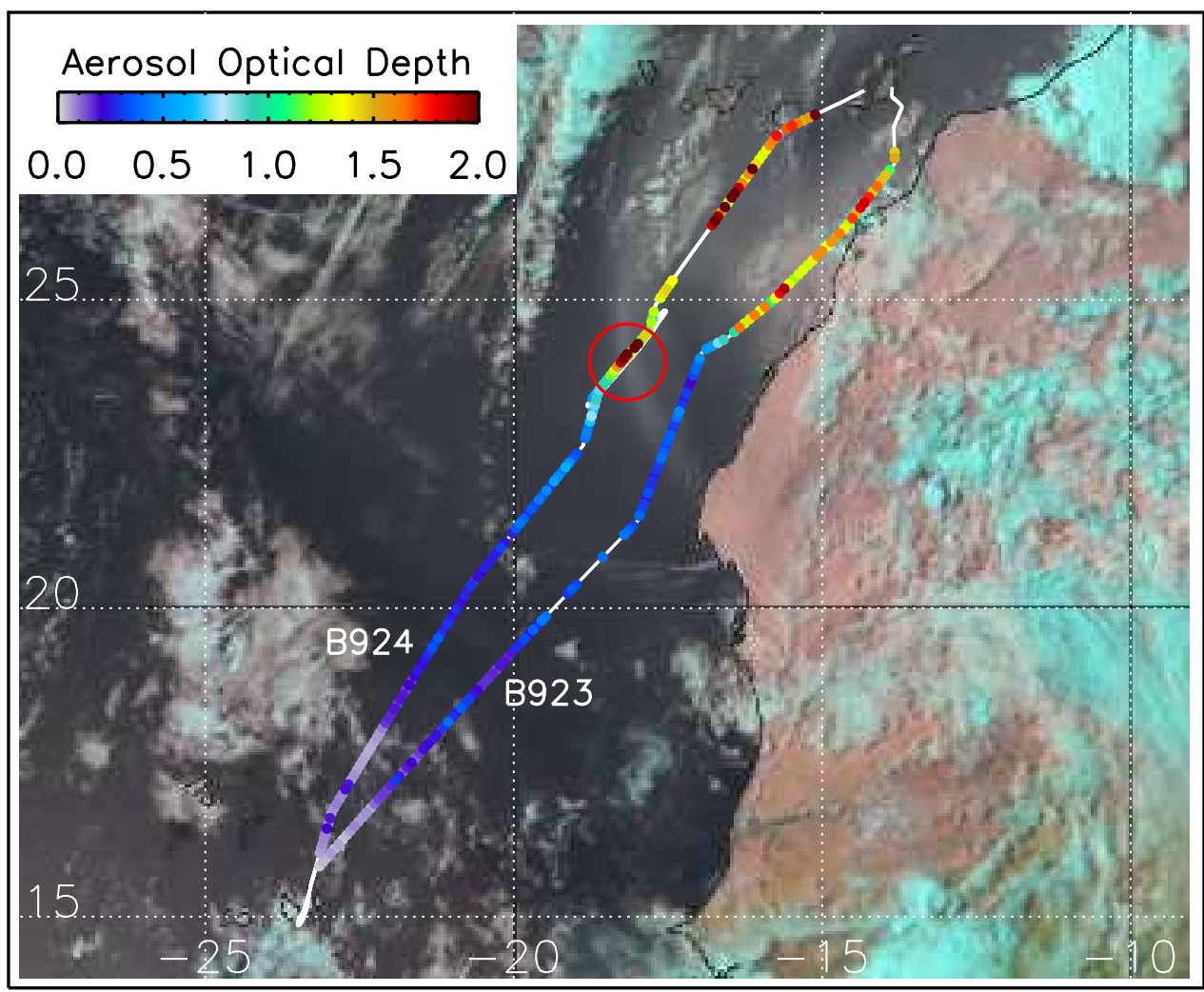

**Figure 3.** Meteosat Second Generation "321 RGB" image for 12 August at 16:30UTC, showing the extent of the dust plume advected off the West coast of Africa. The track of flights B923 and B924 is overplotted. A colour scale depicts the lidar-derived AOD, and a red circle indicates the location where the aircraft descended to sample the heavy dust in situ. A SLR at 30 m altitude was performed between 15:52 and 16:04UTC (run 4), and a SLR at 1,000 m took place between 16:09 and 16:29UTC (run 5).

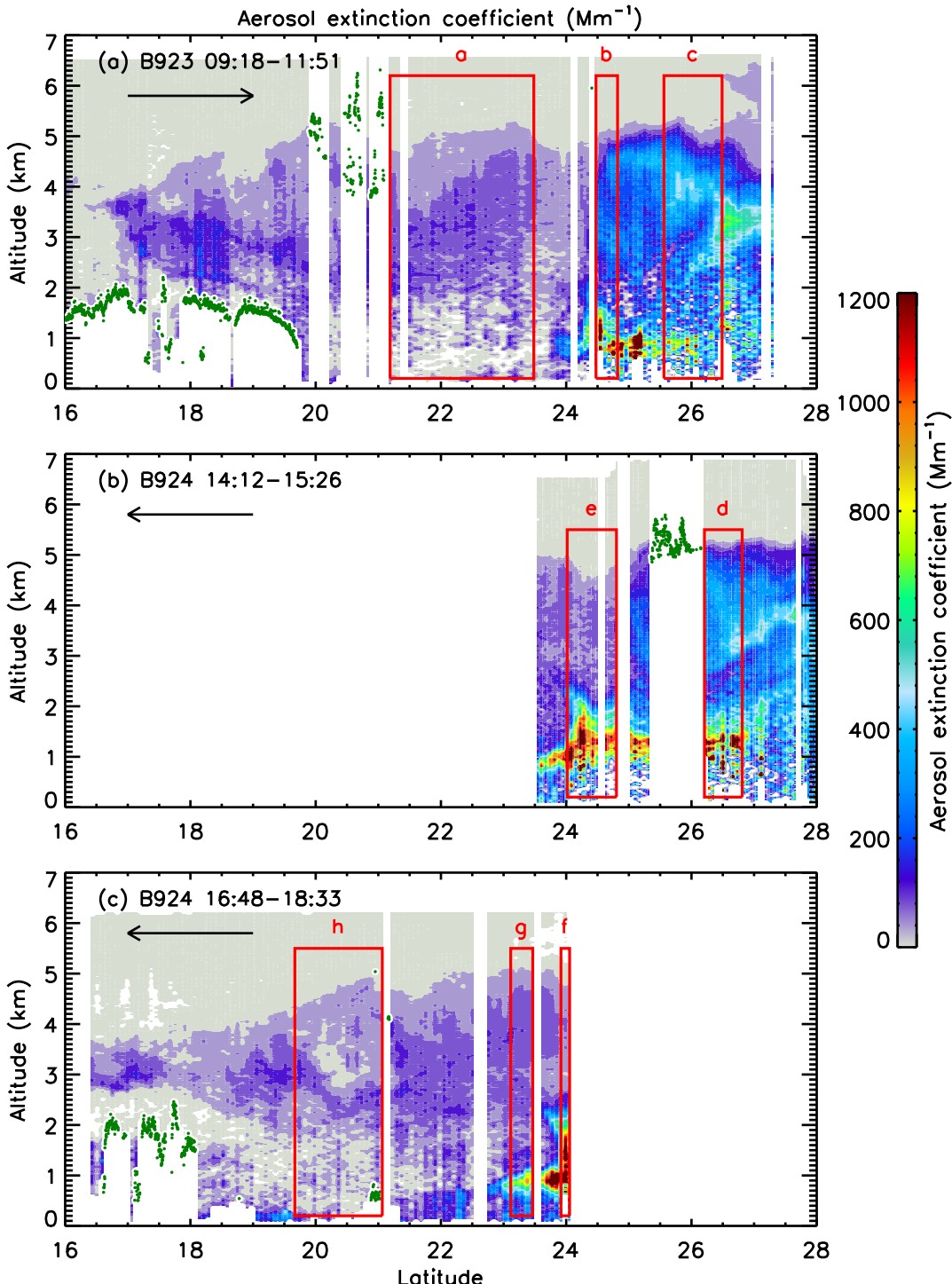

**Figure 4.** Bidimensional structure of the atmosphere on 12 August, for the high altitude transects: (a) flight B923 from 15N to 28.5N; (b) first part of B924, from 28.5N to 23.5N; second part of B924, from 24N to 15N. This curtain plot is coloured according to the aerosol extinction coefficient measured by lidar at 1 min integration time ($\sim$ 9 km footprint). The green dots indicate cloud tops detected with the lidar at 2 s integration time ($\sim$ 300 m horizontal resolution). The red boxes and letters identify the transect portions that have been used to derive the profiles displayed in Fig. 1. The arrows indicate the direction of travel of the aircraft.

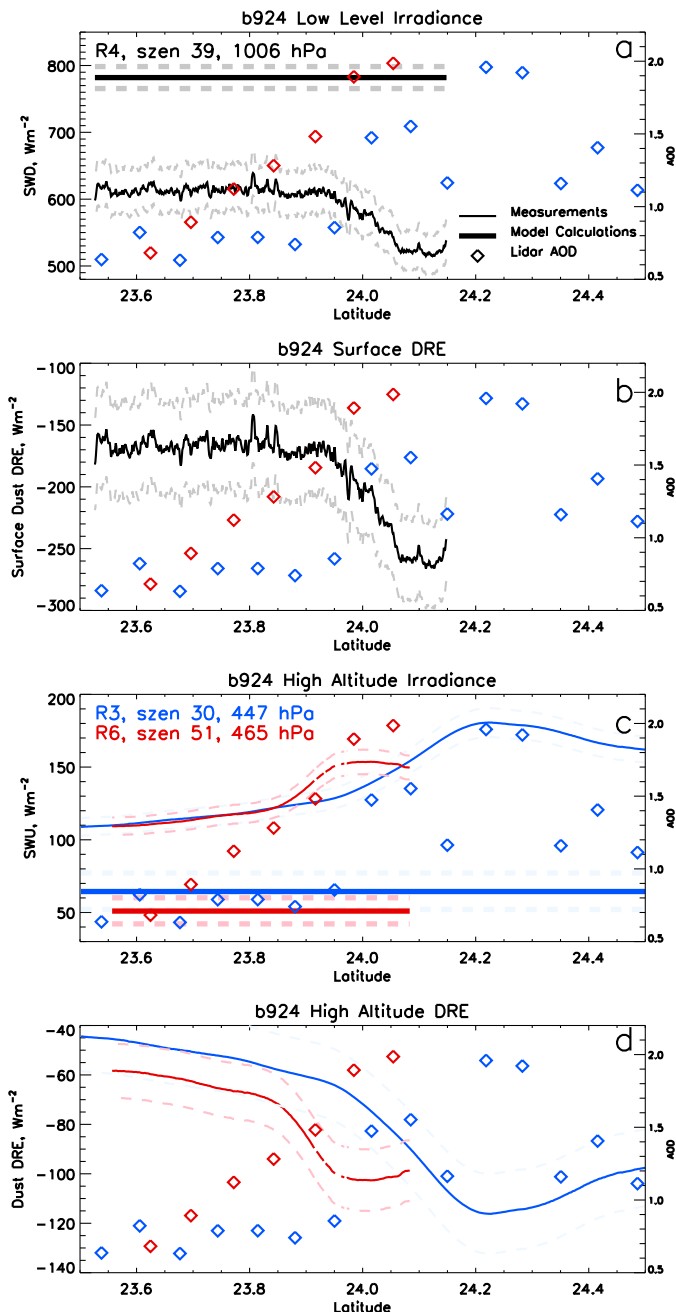

**Figure 5.** Solar irradiance and instantaneous direct radiative effect of dust ahead of and within the dust "front" during flight B924. Observations are lightweight lines, model clear sky calculations are bold lines, uncertainties are displayed through dashed lines, and lidar AODs are indicated with diamonds. (a) Low level (run 4, 30 m) measured and modelled shortwave downwelling irradiance; (b) surface direct radiative effect on the shortwave downwelling irradiance; (c) high altitude shorwave upwelling irradiance for 2 aircraft legs (run 3, blue and run 6, red); (d) high altitude (TOA) direct radiative effect on the shorwave upwelling irradiance for runs 3 and 6.