# Peer review of "Unexpected vertical structure of the Saharan Air Layer and giant dust particles during AER-D"

_Atmospheric Chemistry and Physics, 2018_

## Referee Comment (RC1) · Anonymous Referee #1 · 11 Sep 2018

The manuscript nicely provides an overview on the research flights performed during the AER-D aircraft campaign (6-25 August 2015). The presented study documents the vertical structure of the dust transport layer over the tropical eastern Atlantic between the Cape Verde Islands and the Canary Islands, highlighting a case of an exceptional SAL structure occurring during the AER-D campaign. For this, the authors make use of airborne lidar data, particle size distribution (more details in Ryder et al., 2018), and pyranometer measurements.

General comments:

(1) In the abstract (line 15-16), the authors suggest that "future campaigns should focus more on events with high aerosol load"- Would that create a bias towards high dust loading events? In particular as such events are less frequent than dust events

with average or low dust loadings.

(2) Introduction: It may be worth to add a few more sentences on the seasonal variability of the SAL regarding height, extent, presence etc. in order to seasonally place the results obtained in the framework of AER-D.

(3) The individual section on lidar measurements (vertical distribution of dust particles), dust particle size distribution and pyranometer measurements could be tied together in closer way. Something like a systematic clustering of cases (flights) could be a way to illustrate coincidences as an interpretation guidance on the one side, and be an outcome with potential of application beyond AER-D on the other side.

(4) Page 4, line 19: "exceptional vertical structure". Exceptional with respect to which reference? Please clarify.

(5) Which impact of the meteorological situation / atmospheric circulation regime on the occurrence of the exceptional SAL structure can be expected? Was the atmospheric circulation regime during August 2015 unusual ultimately allowing for the formation of such a dust front? Some sentences on the meteorological / atmospheric situation would help to understand the meteorological circumstances resulting into this case.

(6) Following the present structure of the manuscript, results on lidar measurements, size distribution and radiation are presented in separate sections. The conclusion section briefly addresses all measurement techniques applied in one section. A discussion combining lidar profiles, particle size distribution and pyranometer measurements in concert would provide the opportunity to thoroughly tie together and benefit from this multi-method approach.

(7) Which role play the giant particles for radiative forcing? A brief paragraph discussing this would be a valuable contribution, in particular with respect to the motivation given in the introduction section.

(8) Page 9, line 23-30: Lightning due to presence of dust aerosol or due to meteorological condition? The link is quite interesting, however, here it remains rather speculative. Maybe some more arguments can be provided? (Please see also comment (5) above.)

(9) Page 10, line 6-13: Here, a list of suggestions for further investigations is provided. The first suggestion is on quantifying how unusual the observed vertical structure was by means of satellite observations. The manuscript could benefit from including such a study as this would extend the scope of the results presented. This would also contribute to comment (4) made above as it could serve as a kind of reference when identifying exceptional structures.

Minor comments:

Page 6, line 24: Are times given in UTC? Please clarify.
* * *

---

## Referee Comment (RC2) · Anonymous Referee #2 · 11 Oct 2018

General:

The paper is well written and provides new inside into the microphysical properties of Saharan dust at the beginning of the long range transport across the Atlantic, but still close to Africa. Minor revisions are required.

Details:

Title: The word 'unusual' suggests that the findings clearly deviate from typical findings. And this also implies that the authors measured many cases with ' typical' conditions so that they can conclude: These findings are unusual...! Is that the case? Or does 'unusual' only mean: We did not expect what we found.

P1, L12: There are clear definitions for dust particles and sand particles. Sand particles

have diameters > 60micrometer, smaller ones are dust particles. So, how do you define giant particles?

P1, L12: Latest research on SAL characteristics (lidar based) are presented by Rittmeister et al. (ACP, 2017) and Ansmann et al. (ACP, 2017). Should be cited because they provide some new knowledge on long range transport, removal of dust, mixture of dust with pollution and/or marine particles.

P1, L22: ..may underestimate the size. . .. What does that mean? If possible, provide some more insight! Do you mean. . . of the coarse-mode dust particles, or of the fine-mode dust particles, or is that related to the entire size distribution?

P2, L5: Because this a paper is showing a lot of lidar observations, one should provide more references to SAMUM and SALTRACE aerosol lidar observations (Gross et al., Tellus 2011, ACP 2015, Tesche Tellus 2011, Haarig, ACP 2017).

P2, L6-13: Again, please check the SAL-related papers of Rittmeister et al. (ACP 2017) and Ansmann et al. (ACP 2017) for latest information on dust removal aspects and consequences for the size distribution.

P2, L26: Please check the papers of Tesche et al. (2011a, 2011b in Tellus), and also of Veselovskii et al. (ACP, 2016, Senegal lidar observations).

P4, L25-28: Your observations are made in the near-range of the long-range transport regime, please keep that in mind. The findings are fine! But cannot be taken to make clear statements on ... anything about the microphysics in the Barbados, South America and North America regions....

P4, L26: 'Anomalous' again suggests that in most cases (say in 95% out of all cases) you do not find such structures over the Atlantic. Is that the case? Otherwise, the finding could be denoted as surprising ....

P5, L13: please tell clearly, . . . your write: coarse mode is centered at 5-6 microns (in radius?, diameter?).

P5, L15: Again: fine-mode peaks at 0.25-0.3 microns. . . radius? diameter?

P6, L19: Again: 'giant particles' is not a well defined quantity, better use sand particles, or provide clear diameter boundaries.

P8, L20-25: Again, the observations were performed in the near-range of the long-range transport regime. . . General conclusions (for the entire long range transport regime down to the Americas) cannot be draw.

P10, L8: . . . 200-300 km off the coast of West Africa . . . this statement corroborates that the observations are quite close to the Sahara dust source. . ...., and must thus be carefully discussed, conclusions towards long-range transport consequences cannot be drawn, are just speculative to my opinion.

Check literature: Liu et al, ACPD from 2017, should be ACP now, Mortier et al., 2017. . .. journal? Sequence: Ryder et al., 2018, 2013, 2015 should be Ryder et al. 2013, 2015, 2018. . ., Williams et al, journal?, Yorks et al., journal? All in all: a nice paper and a valuable addition to the dust observation literature!

---

## Author Comment (AC1) · 28 Nov 2018

We would like to thank both reviewers for reading our manuscript carefully, appreciating its value, and bringing a wealth of suggestions. We have followed most of the advice provided, and we believe that as a result the manuscript is much improved. We greatly enjoyed the reviewers' engagement and positive approach.

Hereby we present a point by point response to the reviewers. The manuscript with changes highlighted will indicate the reviewer comment number associated with each change (using PDF notes).

**Anonymous Referee #1**

The manuscript nicely provides an overview on the research flights performed during the AER-D aircraft campaign (6-25 August 2015). The presented study documents the vertical structure of the dust transport layer over the tropical eastern Atlantic between the Cape Verde Islands and the Canary Islands, highlighting a case of an exceptional SAL structure occurring during the AER-D campaign. For this, the authors make use of airborne lidar data, particle size distribution (more details in Ryder et al., 2018), and pyranometer measurements.

We thank the reviewer for carefully reading our paper and providing advice on how to improve it. We believe that the advice in this review is very useful, and contributes to a substantial improvement of the article.

General comments:

**(1)** In the abstract (line 15-16), the authors suggest that "future campaigns should focus more on events with high aerosol load"- Would that create a bias towards high dust loading events? In particular as such events are less frequent than dust events with average or low dust loadings.

Campaign data are by definition sparse and they usually target specific objectives, that investigators choose to study. A base location and a time of the year are chosen, based on existing climatologies, in order to enhance the chances of meeting a certain atmospheric condition (e.g. a dust-laden layer). A set of instrumentation is also chosen, and this decides what will be observable. Weather information is used before and during every flight in order to place the aircraft in the right place. Actively choosing what to measure, when to measure and where to measure permits using the budget and manpower efficiently. Therefore, every campaign, and sometimes every flight, may enhance knowledge about conditions that had not been sampled earlier, or that had been sampled with different instrumentation.

Interpreting campaign data as representing all conditions with equal probability (we think that this is what the reviewer refers to) is not generally a correct assumption, although we acknowledge that it may have been made at times for specific purposes, when better knowledge was missing. Choosing to focus one or more future campaigns on intense events does not mean that a bias will be introduced, but that more knowledge can be generated. This knowledge will require a more in depth interpretation than simply averaging atmospheric properties with previously existing campaigns.

In the current paper, for instance, we report a case with properties that are undocumented in the literature: for flights B923 and B924 we deliberately targeted the heaviest dust loads and chose to place the airplane into them: this required extra

effort, and these peculiar atmospheric conditions would have been missed without explicitly targeting the event. During a fraction of those two flights (note: a fraction), we observed something new. A posteriori, we can say that this flying strategy was a good choice, because it revealed the unprecedented and unexpected vertical structure. We also need to note that positioning the airplane carefully was made possible by the immense progress in numerical weather predictions and satellite observations.

With the sentence highlighted by the referee, we indicate that there may be something to learn when in the future one targets intense events once again: either our current results could be confirmed (or denied!), or possibly new discoveries could be made. Moreover, if this is repeated a number of times, a better idea could be achieved of how frequent (or infrequent) the anomalous vertical structure can be found in such events.

However, we do not say that the observations would be representative of "average conditions", and we think that it would be a mistake to think that campaign data could be interpreted in that sense.

Re-reading the sentence highlighted by the reviewer, however, we think that it sounds too prescriptive, and we shall change the word "should" into "could".

**(2)** Introduction: It may be worth to add a few more sentences on the seasonal variability of the SAL regarding height, extent, presence etc. in order to seasonally place the results obtained in the framework of AER-D.

We shall expand on this in the revised version, based on the papers by Liu et al (2008) and Tsamalis et al (2013) who give a very good description of what the reviewer is asking.

**(3)** The individual section on lidar measurements (vertical distribution of dust particles), dust particle size distribution and pyranometer measurements could be tied together in closer way. Something like a systematic clustering of cases (flights) could be a way to illustrate coincidences as an interpretation guidance on the one side, and be an outcome with potential of application beyond AER-D on the other side.

We do not fully understand what the reviewer suggests to add to the paper, that is not already in it. Please note that results are already clustered as described below.

In the section on vertical structure (see Figure 1), we have the following clusters: (a) moderate dust, most often encountered in the campaign, and in agreement with the conventional model and (b) heavy dust, as observed on 12 August, and unusual.

In the section on particle size distribution, we cluster the following samples: marine boundary layer (blue in Figure 2), moderate dust (red), and heavy dust (green). The heavy dust PSD corresponds to the same location of the heavy dust vertical profile, and the moderate dust PSDs correspond to all the other in situ samplings in the SAL, corresponding broadly to the moderate dust vertical profiles (i.e. the clustering is consistent with the previous section). The MBL PSDs correspond to samplings at 30 m above sea level, and below the SAL.

The section on pyranometer measurements only describes the 12 August, and we present measurements South of the dust front (i.e. in moderate dust) and North of the front (i.e. in heavy dust). As the main factor affecting the shortwave radiation is the

solar downwelling, it would overcomplicate the picture to show all the flights here, as we would have to disentangle the effect of the solar zenith angle. Moreover, in some of the other cases the picture may also be affected by the presence of clouds nearby. We believe therefore that it is best to stick to our approach to only show flight B924 in section 6.

To help make things clearer, we shall clarify in section 4 that the clustering is consistent with section 3, and in section 6 that the Southern end of the plots correspond to "moderate dust" and the Northern end to "heavy dust".

**(4)** Page 4, line 19: "exceptional vertical structure". Exceptional with respect to which reference? Please clarify.

We agree with the reviewer that, although the vertical structure was "exceptional", here it is probably better to write "different" and move the word "exceptional" to the last sentence of the paragraph, where an explanation is given (i.e.: not previously encountered over the Atlantic).

**(5)** Which impact of the meteorological situation / atmospheric circulation regime on the occurrence of the exceptional SAL structure can be expected? Was the atmospheric circulation regime during August 2015 unusual ultimately allowing for the formation of such a dust front? Some sentences on the meteorological / atmospheric situation would help to understand the meteorological circumstances resulting into this case.

We shall add a description of the meteorological situation in August 2015 in section 2.

Please note, however, that we have no evidence pointing towards an anomalous wind regime during this month. In other words, we think that the evidence for an anomaly is limited to the 12-13 August, and it resides in the observations that of (1) an intense event with an unprecedented vertical structure; and (2) an anomalous lightning activity, both of which we report.

**(6)** Following the present structure of the manuscript, results on lidar measurements, size distribution and radiation are presented in separate sections. The conclusion section briefly addresses all measurement techniques applied in one section. A discussion combining lidar profiles, particle size distribution and pyranometer measurements in concert would provide the opportunity to thoroughly tie together and benefit from this multi-method approach.

Sections 3 and 4 discuss the vertical structure and the particle size spectrum; in both of them the heavy dust measurements stand out. Section 5 explains better this event, explaining in particular how the measurements have been taken, and how they fit in a wider picture of a dust outbreak and the advance of a dust front. Then, section 6 explores the radiative effect before the dust front (moderate dust) and after it (heavy dust). As the reviewer points out, the conclusions put together the results from sections 3-6, before opening to the scientific questions that our study raises. We believe that there is therefore no need to add an additional section to tie together the observations.

As already mentioned under (3) above, we will better clarify in sections 3, 4 and 6 how the clustering of the data works between the moderate and heavy dust conditions, and we shall make sure that we consistently apply these two terms throughout the

paper. We will also repeat this clustering in two dust conditions in the conclusions, hoping therefore to bring better clarity.

**(7)** Which role play the giant particles for radiative forcing? A brief paragraph discussing this would be a valuable contribution, in particular with respect to the motivation given in the introduction section.

We will add such a paragraph to our conclusions.

**(8)** Page 9, line 23-30: Lightning due to presence of dust aerosol or due to meteorological condition? The link is quite interesting, however, here it remains rather speculative. Maybe some more arguments can be provided? (Please see also comment (5) above.)

The reviewer is right: the link between dust and electric activity is speculative, and it is beyond the scope of this paper to try and demonstrate a causal dependency. As we mentioned in the article, the understanding of the link between aerosols and lightning discharges is still weak and contradictory. We however believe that it may be useful to highlight this coincidence, as it may be useful for further studies.

We will clarify better that we do not claim to prove a causal effect.

**(9)** Page 10, line 6-13: Here, a list of suggestions for further investigations is provided. The first suggestion is on quantifying how unusual the observed vertical structure was by means of satellite observations. The manuscript could benefit from including such a study as this would extend the scope of the results presented. This would also contribute to comment (4) made above as it could serve as a kind of reference when identifying exceptional structures.

Although we believe that a systematic satellite-based study of the dust vertical structure could bring information on how frequently the anomalous vertical structure can be encountered (using passive and/or active sensors), we believe that performing such a study is not trivial, and would be well beyond the scope of the current paper.

**(10)** Minor comments: Page 6, line 24: Are times given in UTC? Please clarify.

All times in the paper are UTC (see e.g. the figure captions). We will clarify this by adding a sentence at the end of the section on the Research flights, and by specifying UTC in the sentence highlighted by the reviewer and a few other places in the article.

**Anonymous Referee #2**

General: The paper is well written and provides new inside into the microphysical properties of Saharan dust at the beginning of the long range transport across the Atlantic, but still close to Africa. Minor revisions are required.

We thank the reviewer for carefully reading our paper and for his or her appreciation. We have found the advice very constructive and we address it in the following, with the result of an overall substantial improvement of the manuscript.

*Note: for practical reasons we have added numbering to the reviewer's comments below, continuing from the numbering initiated by reviewer #1.*

**(11)** Title: The word 'unusual' suggests that the findings clearly deviate from typical findings. And this also implies that the authors measured many cases with ' typical' conditions so that they can conclude: These findings are unusual: : :! Is that the case? Or does 'unusual' only mean: We did not expect what we found.

The "typical" structure is documented in the literature, and this is explained in the introduction, where a conceptual model is outlined, and several references addressing the vertical distribution of dust in the Saharan Air Layer have been cited (Karyampudi et al, 1999; Liu et al, 2008; Tsamalis et al, 2013; Senghor et al, 2017; etc.). In the current paper we report six flights: therefore our own measurements are sporadic.

As reported in the section on vertical structure, describing our observations "in most cases, a deep dust layer is identified, with base at 1–2 km and top at 5–6 km altitude, above a MBL also displaying a significant aerosol content. [...] This observed structure is in agreement with expectations from the conventional model for Saharan dust transport over the Atlantic (Carlson and Prospero, 1972; Karyampudi et al., 1999)."

Therefore "unusual" here means first of all "we did not expect what we found" (i.e. our findings show a case where a new vertical structure is highlighted, not previously observed). It also means "within the context of our few measurements, these findings are a deviation from what we saw in a majority of cases".

However "unexpected" may as well work to describe this, and we will substitute this word in the title. Thank you for suggesting it.

**(12)** P1, L12: There are clear definitions for dust particles and sand particles. Sand particles have diameters > 60micrometer, smaller ones are dust particles. So, how do you define giant particles?

We thank the reviewer for raising this point. The term "sand" is more often used in the vicinity of a desert area, whereas the term "dust" can denote the aerosol lifted at a high altitude and/or transported over a distance (see e.g. https://www.accuweather.com/en/home-garden-articles/earth-you/sandstorms-and-dust-storms-and/32499921; https://en.wikipedia.org/wiki/Dust_storm). Kok et al report that "In the geological sciences, sand is defined as mineral (i.e., rock-derived) particles with diameters between 62.5 and 2,000 um , whereas dust is defined as particles with diameters smaller than 62.5 um (note that the boundary of 62.5 um differs somewhat between particle size classification schemes, see Shao 2008, p. 119). In the atmospheric sciences, dust is usually defined as the material that can be readily suspended by wind (e.g., Shao 2008), whereas sand is rarely suspended and can thus form sand dunes and ripples, which are collectively termed bedforms." (https://arxiv.org/pdf/1201.4353.pdf).

In light of the above, we prefer to maintain the term "dust" as used in the atmospheric sciences. For clarity, we will add the following sentence in the section of the article on particle size distribution: "We note that authors in the geological sciences often consider that 62.5-2,000 um particles are sand as opposed to dust. Here, however, we will use the term dust for the particles that we observed, adhering to a terminology in use in the atmospheric sciences, where dust is considered to be suspended material transported by the wind (Kok et al, 2012)."

**(13)** P1, L12: Latest research on SAL characteristics (lidar based) are presented by Rittmeister et al. (ACP, 2017) and Ansmann et al. (ACP, 2017). Should be cited

because they provide some new knowledge on long range transport, removal of dust, mixture of dust with pollution and/or marine particles.

Thank you for pointing these out. We will highlight the contribution from these papers in our introduction. Note that we had already referenced the R/V Meteor cruise experiment by citing Kanitz et al (2014) earlier in the introduction; as Rittmeister et al (2017) supersedes that paper, we shall cite the latest one instead.

**(14)** P1, L22: ..may underestimate the size: : :. What does that mean? If possible, provide some more insight! Do you mean: : : of the coarse-mode dust particles, or of the finemode dust particles, or is that related to the entire size distribution?

Kok et al found that atmospheric dust is substantially coarser than represented in current global climate models, i.e. that the particle size distribution in the models has too many fine particles and too few coarse particles. We will rephrase this in our paper to clarify it.

**(15)** P2, L5: Because this a paper is showing a lot of lidar observations, one should provide more references to SAMUM and SALTRACE aerosol lidar observations (Gross et al., Tellus 2011, ACP 2015, Tesche Tellus 2011, Haarig, ACP 2017).

We are surprised if the reviewer feels this way! Papers about SAMUM and SALTRACE and about lidar measurements are already very well represented in this list of experiments on dust (Heintzenberg, 2009; Weinzierl et al, 2009; Ansmann et al, 2011; Chouza et al, 2016; Weinzierl et al, 2017). The additional papers that the reviewer suggests are excellent and very important articles, but we feel that adding more citations here in this list could create in the reader an idea of imbalance in the representation of these two campaigns with respect to other dust campaigns (AMMA, DODO, DABEX, GERBILS, PRIDE, SHADE, NAMMA, CV-DUST, the R/V Meteor transatlantic cruise, and we may possibly be forgetting some). With the purpose of remaining balanced, we shall therefore choose not to follow the reviewer's advice, but we thank him or her in any case for prompting us to double check this list of citations.

**(16)** P2, L6-13: Again, please check the SAL-related papers of Rittmeister et al. (ACP 2017) and Ansmann et al. (ACP 2017) for latest information on dust removal aspects and consequences for the size distribution.

As per comment (13) above, we will reference these papers. Thanks for pointing them out.

**(17)** P2, L26: Please check the papers of Tesche et al. (2011a, 2011b in Tellus), and also of Veselovskii et al. (ACP, 2016, Senegal lidar observations).

We will bring reference to these papers.

**(18)** P4, L25-28: Your observations are made in the near-range of the long-range transport regime, please keep that in mind. The findings are fine! But cannot be taken to make clear statements on ... anything about the microphysics in the Barbados, South America and North America regions....

We thank the reviewer for this comment, with which we fully agree. We shall add a clarifying sentence, indicating the scope of this particular observation.

**(19)** P4, L26: 'Anomalous' again suggests that in most cases (say in 95% out of all cases) you do not find such structures over the Atlantic. Is that the case? Otherwise, the finding could be denoted as surprising ....

The reviewer is correct: in all the other cases, including vertical profiles measured further away from the dust front on that same day, we did not find the anomalous structure. Moreover, we haven not found previous articles documenting it. See also our response given earlier concerning the title. We are in any case happy to use the word "suprising" in this sentence.

**(20)** P5, L13: please tell clearly, : : : your write: coarse mode is centered at 5-6 microns (in radius?, diameter?).
P5, L15: Again: fine-mode peaks at 0.25-0.3 microns: : : radius? Diameter?

All particles sizes are expressed in terms of diameter. We will clarify this in the revised manuscript.

**(21)** P6, L19: Again: 'giant particles' is not a well defined quantity, better use sand particles, or provide clear diameter boundaries.

Please see our response earlier concerning the use of the terms "sand" and "dust". Earlier in the manuscript (page 5 line 14 of discussion paper), we indicated that we define giant particles as those that are 20-80 um in diameter. This indication should avoid the potential ambiguity.

**(22)** P8, L20-25: Again, the observations were performed in the near-range of the longrange transport regime: : : General conclusions (for the entire long range transport regime down to the Americas) cannot be draw.

We understand what the reviewer is saying and we shall substitute the word "ubiquitous" in this sentence with "in the Eastern Atlantic".

**(23)** P10, L8: : : : 200-300 km off the coast of West Africa : : : this statement corroborates that the observations are quite close to the Sahara dust source : : :.., and must thus be carefully discussed, conclusions towards long-range transport consequences cannot be drawn, are just speculative to my opinion.

200-300 km from the coast may still mean 2–3,000 km from the sources; therefore we believe that we remain within the field of long-range transport. However, we do not aim at taking conclusions towards longer transport ranges across the Atlantic: if we have given that impression, definitely we have been incorrect. We hope that the several clarifications emerging from the reviewer's suggestions, addressed in previous points, will remove this false impression. Thanks very much for bringing this risk to our attention.

**(24)** Check literature: Liu et al, ACPD from 2017, should be ACP now, Mortier et al., 2017: : :. journal? Sequence: Ryder et al., 2018, 2013, 2015 should be Ryder et al. 2013, 2015, 2018: : :, Williams et al, journal?, Yorks et al., journal?

Thanks for these; they will all be corrected. Note: the ordering of the articles in the bibliography is controlled by the Copernicus LaTeX style file, and therefore we cannot change the sequence of the Ryder papers. This will be reviewed by the typesetter at time of publication, and they shall be able to order the papers according to the journal's standards.

All in all: a nice paper and a valuable addition to the dust observation literature!

Many thanks for your encouragement!